# Direct Implantation of Patient Brain Tumor Cells into Matching Locations in Mouse Brains for Patient-Derived Orthotopic Xenograft Model Development

**DOI:** 10.3390/cancers16091716

**Published:** 2024-04-28

**Authors:** Lin Qi, Patricia Baxter, Mari Kogiso, Huiyuan Zhang, Frank K. Braun, Holly Lindsay, Sibo Zhao, Sophie Xiao, Aalaa Sanad Abdallah, Milagros Suarez, Zilu Huang, Wan Yee Teo, Litian Yu, Xiumei Zhao, Zhigang Liu, Yulun Huang, Jack M. Su, Tsz-Kwong Man, Ching C. Lau, Laszlo Perlaky, Yuchen Du, Xiao-Nan Li

**Affiliations:** 1Shenzhen Key Laboratory for Systems Medicine in Inflammatory Diseases, School of Medicine, Sun Yat-sen University, Shenzhen 510080, China; qilin23@mail.sysu.edu.cn; 2Texas Children’s Cancer Center, Texas Children’s Hospital, Baylor College of Medicine, Houston, TX 77030, USA; pabaxter@texaschildrens.org (P.B.); mari.kogiso@gmail.com (M.K.); yongganhei2years@gmail.com (H.Z.); fkbraun@outlook.com (F.K.B.); hblindsa@texaschildrens.org (H.L.); sibo.zhao@cookchildrens.org (S.Z.); wan-yee.teo@duke-nus.edu.sg (W.Y.T.); litianyu99118@163.com (L.Y.); zxm616@hotmail.com (X.Z.); zhigangliu1983@hotmail.com (Z.L.); huangyulun@suda.edu.cn (Y.H.); jmsu@texaschildrens.org (J.M.S.); ctman@txch.org (T.-K.M.); clau@connecticutchildrens.org (C.C.L.); naturalperl@gmail.com (L.P.); 3Laboratory of Molecular Neuro-Oncology, Texas Children’s Hospital, Baylor College of Medicine, Houston, TX 77030, USA; 4Ann & Robert H. Lurie Children’s Hospital of Chicago, Department of Pediatrics, Northwestern University Feinberg School of Medicine, Chicago, IL 60611, USA; sxiao@luriechildrens.org (S.X.); aalaaabdallah2023@u.northwestern.edu (A.S.A.); msuarezpalacios@luriechildrens.org (M.S.); ziluhuang@luriechildrens.org (Z.H.); 5The Laboratory of Pediatric Brain Tumor Research Office, SingHealth Duke-NUS Academic Medical Center, Singapore 169856, Singapore; 6Robert H. Lurie Comprehensive Cancer Center, Northwestern University, Chicago, IL 60611, USA

**Keywords:** orthotopic xenograft model, brain tumor, cancer

## Abstract

**Simple Summary:**

In this study, researchers tackled the challenge of advancing therapies for malignant brain tumors, given the scarcity of clinically relevant and biologically accurate mouse models. They introduced a novel surgical technique for transplanting fresh human brain tumor samples into SCID mice, accurately mimicking the original tumor’s location in the brain. Through this method, they successfully established 188 patient-derived orthotopic xenograft (PDOX) models from 408 brain tumor samples, preserving the histopathological and genetic traits of the original tumors. Success rates varied among tumor types, with high-grade glioma demonstrating the highest success rate. Overall, this technique presents a straightforward and effective approach for generating extensive cohorts of tumor-bearing mice for both biological investigations and preclinical drug evaluations, eliminating the necessity for a stereotactic frame.

**Abstract:**

**Background:** Despite multimodality therapies, the prognosis of patients with malignant brain tumors remains extremely poor. One of the major obstacles that hinders development of effective therapies is the limited availability of clinically relevant and biologically accurate (CRBA) mouse models. **Methods:** We have developed a freehand surgical technique that allows for rapid and safe injection of fresh human brain tumor specimens directly into the matching locations (cerebrum, cerebellum, or brainstem) in the brains of SCID mice. **Results:** Using this technique, we successfully developed 188 PDOX models from 408 brain tumor patient samples (both high-and low-grade) with a success rate of 72.3% in high-grade glioma, 64.2% in medulloblastoma, 50% in ATRT, 33.8% in ependymoma, and 11.6% in low-grade gliomas. Detailed characterization confirmed their replication of the histopathological and genetic abnormalities of the original patient tumors. **Conclusions:** The protocol is easy to follow, without a sterotactic frame, in order to generate large cohorts of tumor-bearing mice to meet the needs of biological studies and preclinical drug testing.

## 1. Introduction

Deaths caused by central nervous system (CNS) tumors result in the highest mortality among pediatric and adult cancers. Over the past thirty years, there have been only modest incremental improvements in the outcomes of children with high-grade CNS tumors. Similar to adult glioblastoma (GBM) tumors, pediatric brain tumors—particularly high-grade gliomas (HGG), diffuse intrinsic pontine gliomas (DIPG), and recurrent medulloblastomas (MB)—and ependymomas have a dismal prognosis, and survivors of these tumors often experience devastating treatment-induced toxicities including neurocognitive and endocrine sequelae [1,2,3,4,5]. Consequently, clinically relevant models are needed to understand tumor biology and to develop more effective and less toxic therapeutic options. Additionally, multiple seminal studies have revealed complex biological heterogeneities of brain tumors and identified molecularly distinct subgroups within the classic pathological diagnosis of brain tumors [6,7,8,9,10,11,12,13,14,15,16,17,18,19,20,21,22]. These findings highlight the need to develop a large panel of animal models replicating the full spectrum of molecular subtypes of pediatric brain tumors.

In addition to creating genetically engineered mouse (GEM) models, advances have been made in developing patient-derived xenograft (PDX) mouse models of various human cancers [23,24,25,26,27,28,29]. Some of the PDX tumors are still maintained in the subcutaneous compartment, making it difficult to faithfully replicate the tumor microenvironment and potential alteration of tumor biology. The subcutaneous PDX models do have an advantage of direct visual inspection and easy measurement of tumor size changes. Since brain tumor biology and behavior are critically dictated by the unique environment of different areas of the brain and the blood–brain barrier plays a critically role in determining drug delivery and the overall efficacy of novel therapies [30,31], it is highly desired for brain tumor models to replicate the exclusive microenvironment of human brains [32].

**GOAL:** For the past 20 years, our laboratory has been actively engaged in the development of animal models for pediatric brain tumors through direct implantation of patient tumor cells into the matching locations in the brains of SCID mice. Our goal is to develop a panel of patient-derived orthotopic xenograft (PDOX or orthotopic PDX) mouse models of brain tumors that is ***c***linically ***r***elevant and ***b***iologically ***a***ccurate (**CRBA**). Ideally, such CRBA PDOX models (1) are patient-specific, replicating the original tumor’s histopathological features, invasive/metastatic phenotype, and cancer stem cell pool; (2) grow in a microenvironment that shares maximum similarities with their natural habitat; (3) preserve the key genetic and epigenetic abnormalities of the original patient tumors; (4) represent different clinical stages of brain tumors, including treatment-naïve at diagnosis, recurrence, and terminal/lethal; (5) cover a broad, if not whole, spectrum of molecular subtypes of pediatric brain tumors, e.g., including four groups (SHH, WNT, group 3 and 4) of medulloblastomas (MB); (6) have multiple models from each molecular subtype to reproduce the inter-tumoral heterogeneity of each subtype; and (7) can be produced in large cohorts (>150–200/day) and on demand to meet the needs of complex in vivo drug testing and biological studies.

To achieve this goal, we have optimized a surgical protocol that allows for rapid and safe implantation of patient brain tumor cells into the matched locations in the brains of SCID mice, i.e., cerebral tumors to mouse cerebra, cerebellar tumors to mouse cerebella, and brain stem tumors to mouse brain stem [33,34,35,36,37,38,39]. Using this protocol, we have established 188 patient-derived orthotopic xenograft models (PDOX or orthotopic PDX) from 13 different types of brain tumor pathologies (Table 1). Since brain tumors are relatively rare and many of them, particularly pediatric brain tumors, have been (and are still being) further subclassified into multiple molecular groups, we wish to share our experience with the development of the PDOX model to facilitate the establishment of CRBA PDOX models that represent the full spectrum of molecular subtypes of all pediatric and adult brain tumors.

**Application:** This technique can be broadly applied to all types of brain tumors, including cerebral, cerebellar and brain stem tumors; malignant and benign tumors; both pediatric and adult brain cancers; primary CNS tumors; and metastatic brain cancers. All brain cancer investigators either on tumor biology or preclinical drug testing should find the method of interest and helpful for their studies.

**Advantages of this strategy:** (1) *Rapid:* implantation of one mouse takes <60 s, starting from hair removal to the withdrawal of injection needles. A team of three investigators can implant 150–200 mice in 4 h. (2) *Safe:* the surgically related death rate is less than 0.5% after ~40,000 orthotopic implantations; (3) *Precise depth of tumor cell engraftment*: the depth is determined by a stopper attached to a Hamilton gas tight needle at a pre-determined depth. (4) *Highly reproducible*: the same location, same depth, and same cell number are easily achieved in any given cohort of animals. (5) *Only a small amount of tumor cells are needed:* as few as 0.1 million patient tumor cells are needed per mouse compared with 1–2 million cells for subcutaneous engraftment. (6) *High tumor take rate:* a tumor formation rate greater than 70% is achieved in GBM, and >60% for medulloblastoma; even tumors of lower-grade malignance form orthotopic xenografts. (7) *No need for extra-equipment*: this a freehand procedure, the location of blur hole can be easily set, and the depth of tumor cell injection is fixed by a stopper. (8) *Biologically accurate*: brain tumors can be grown in corresponding locations in the mouse brain, which share maximum similarities with tumors’ natural habitat and the critical microenvironment due to the preservation of key biological features and genetic/epigenetic abnormalities. (9) *Clinically relevant*: the existence of the blood–brain barrier enables the evaluation of the efficacy of in vivo drug delivery and the overall therapeutic efficacy. (10) *Remarkably increased success in developing matching in vivo models*: compared to the difficulties of obtaining patient tissues and the low success rate of in vitro growth of brain tumor cells, PDOX tumors provide a resourceful platform for establishing matching panels of in vitro models (monolayer, 3D neurosphere and organoids).

**Limitations:** Training and practice are needed. This is still a procedure used on animal brains. After initial training, more practice will rapidly improve the efficiency and the accuracy (in terms of location) of the model’s establishment.

## 2. Experimental Design

This protocol details the procedure of the freehand transplantation of surgical specimens of brain tumors into matching locations in the brains of SCID mice. By implanting human brain tumor cells directly into matching locations in the brains of SCID mice, these PDOX mouse models may more closely replicate the microenvironment of brain tumors, thereby overcoming many disadvantages associated with subcutaneous xenografts. Initiating these models from fresh surgical specimens and performing serial sub-transplantations strictly in vivo in mouse brains avoids the need for archaic cell lines. In addition, these models provide an ongoing supply of fresh tumors cells for the subsequent establishment of in vitro models (monolayer, neurosphere and organoid) for pre-clinical testing [34,37,40]. In summary, direct orthotopic transplantation of patients’ tumor tissues has been shown to better replicate the molecular, cellular, and clinical phenotypes of the original tumor [33,36,41,42,43].

Traditional intra-cranial injection of tumor cells often uses a stereotaxic frame. While this technique provides precise positioning of injected tumor cells, the process is laborious and time-consuming and can limit the output of animal models. Here, we describe a protocol of “freehand” injection, the safety, reproducibility, and a high rate of initial tumor engraftment of which have been previously demonstrated in different types of pediatric brain tumors [33,37,44]. Through the use of this freehand injection technique, we are able to create large groups of tumor-bearing mice to meet the demands of pre-clinical testing. An experienced team of two or three investigators can inject 30 to 50 mice per hour. Even after repeated serial sub-transplantations of xenograft tissue, the genetic and phenotypic characteristics of the tumor are well maintained [45,46,47]. The preservation and expansion of these tumors, as well as the ability to develop large cohorts, highlight the fidelity and utility of this protocol and our orthotopic xenograft models.

## 3. Materials and Methods

**A.** 
**Animals**
Mouse strains: Three strains of SCID mice, (1) Rag2/SCID, (2) NOD/SCID (NOD.^129S7(B6)-Rag1tm1Mom/J^) and NSG (Jax Lab), housed in a specific pathogen-free animal facility.
○An advantage of NOD/SCID (NOD.^129S7(B6)-Rag1tm1Mom/J^) is that these mice are resistant to radiation and can tolerate fractionated ionizing radiation (2 Gy/day × 5 days) [48].Animal age: To replicate the developing brain of pediatric brain tumors, younger mice (5–8 weeks) are preferred. For adult brain tumor models, mice older than 10 weeks (10–14 weeks) are usually used.Standardizing animal age can also reduce variabilities in animal body weight and head size. The latter is also important for minimizing the differences in the depth of tumor implantation.Animal sex: Both male and female mice were used, although no significant differences in tumorigenicity, growth rate, and drug responses were observed as we reported previously [37].
**B.** 
**Model establishment from patient tumors**


All experiments should be conducted following an Institutional Animal Care and Use Committee (IACUC)-approved protocol.
*B.1. Reagents, Equipment and Supplies*Reagents

The following is a list of reagents used by our laboratory. This can be modified according to the users’ specifications or vendor preference, unless otherwise noted:Media: DMEM (Cellgro 10-013-CV) with 10% Fetal Bovine Serum (Cellgro 35-010-CF)10×PBS (Cellgro 46-013-CM)Sodium pentobarbital (50 mg/mL) (Abbott NDC 67386-501-52)Sterile water (Abraxis 401753D NDC 63323-185-10)Trypan blue solution (Sigma T8154)Beuthanasia-D special (100 µL/15–20 g)
Equipment and Supplies
Rodent Anesthesia Machine, Cat #. VAS 2007R, Veterinary Anesthesia Systems, Inc., Phoenix, AZ, USA.High-speed surgical drill
(Fine Scientific Tools, 18000-17, which may have been discontinued)Ideal Micro-Drill™ (CellPoint Scientific, Gaithersburg, MD, USA. http://www.cellpointscientific.com/Products/Ideal-Micro-Drill/67-1200A, accessed on 1 December 2012)
Micro-drill stainless steel burrs—0.7 mm (Pack of 10) (Fine Scientific Tools, Foster City, CA, USA)Petri dish 100 × 15 cm (VWR 25384-302)15 mL conical tube (Cellstar 188271)50 mL conical tube (Cellstar)10-blade scalpel (Tyco 131610)Eppendorf tube (1.5 mL)Hamilton 10 µL Gastight syringe 1701 attached with a custom-made stopper (Hamilton Company, Timis County, Romania)Tissue adhesive (Tissumed II synthetic absorbable tissue adhesive, Cat.NO 3002931, Veterinary Products Laboratories, Miami Lakes, FL, USA)Scissors100 µm and 40 µm strainer (Falcon Ref 352340/Ref 352340)Alcohol wipes
*B.2. Procedure*

Preparation of fresh patient tumors should be completed in a biosafety hood, using universal precautions. Individual institutional review board (IRB) approval should be obtained prior to the use of patient specimens. Preparation of tumor tissues from fresh specimens for transplantation proceeds as follows:Fresh tumor tissues are received from the pathology laboratory and should be processed immediately whenever possible.
○*A unique tumor ID should be created and shared with all the stakeholders (neurosurgeons, pathologist, oncologist, tumor bank, etc.).*○*It is desired that tumor sample processing be completed within 60 min after resection to maximize the preservation of cell viability.*The specimen is transferred with fresh cold (4 °C) media (DMEM + 10% FBS).The tumor is cut into small pieces in a sterile Petri dish with scissors and dissociated into single cells.
○*Malignant brain tumor tissues are usually very soft and easy to mechanically dissociate.*○*Enzyme digestion with trypsin or collagenase/halogenase can also be applied with optimization.*The cell mixture is pipetted up and down for further dissociation.Cells are filtered through a 100 µM and 40 µM strainer into a 50 mL conical tube to collect single cells.
○*Note: Small clumps of tumor cells (~5 cells) are occasionally seen after filtration. These spheroids rarely interfere with the injection as they can pass through the needles easily.*○*One added advantage of such small clumps is that the center cells can potentially be well protected by the outer layer of cells and stay free of mechanical damage.*
The Petri dish is washed once with PBS to ensure all cells are obtained and filtered again.Cells are counted using trypan blue.Cells are spun at 200 g for 5 min and media are removed.Cells are resuspended in DMEM+ 10% FBS media that is pre-cooled to 4 °C for a final cell count of 5 × 10^4^ per microliter.Cells are transported on ice to the animal facility.As an alternative option: Ideally, tumor cells should be transplanted while fresh. If a delay is inevitable, cells can be temporarily stored in liquid nitrogen using DMEM media supplemented with 20% FBS and 10% DMSO (however, freezing tumor cells in liquid nitrogen will decrease the percentage of viable cells).


**C.** **Transplant:** Brain tumor cells will be transplanted into the matching anatomical location of the original patient tumor in the mouse brains, i.e., cerebral tumors into mouse cerebra, cerebellar tumors into mouse cerebella, and brain stem tumors into mouse brain stem.
Animal strain: Rag2/SCID, NOD/SCID, and NSG mice are bred and housed per institutional protocols in the animal facility.Pain medication before and/or after tumor cell implantation should be given following an IACUC-approved protocol.
○*Subcutaneous injection of buprenorphine SR (1 mg/kg), which is a sustained release formulation of buprenorphine, 30–60 min prior to tumor implantation provides 48–72 h of analgesia. It allows for more consistent drug plasma levels and decreases the stress of handling associated with repeated injections.*Mice are anesthetized with isoflurane inhalation (Rodent Anesthesia Machine, Cat #. VAS 2007R, Veterinary Anesthesia Systems, Inc.)
○*Alternatively, intraperitoneal (i.p.) injection of sodium pentobarbital (50 mg/kg) may be used.*With the animal deeply sedated, its hair is removed using either of or a combination of the following methods:○*The hair can be neatly shaved using a power cordless trimmer.*○*Depilatory creams (such as Nair®, Veet®, etc.) have been proven to be effective, atraumatic, and non-toxic. The cream should be applied for 30 s, followed by washing and cleaning.*
The animal is held in a prone position and the head secured between the thumb and index finger (Figure 1 and Figure 2A–C).The shaved cranium and the gloved fingers are wiped twice with iodine and once with alcohol.○*Applying an incise drape with iodine (3M) or other sterile drape over the mouse cranium and the holding fingers will further reduce the risk of infection.*
A small (1 mm) skin incision is made to the right of the sagittal sinus and 2 mm anterior (for intra-cerebral injections) (Figure 2E) or 1 mm posterior (for intra-cerebellar injections) (Figure 2I) to the bregma occipital line or at the upper right corner between the sagittal sinus and the bregma occipital line (Figure 2M).○*The sagittal sinus and bregma occipital line are formed by blood vessels fixed on the cranial bone, which are visible as a dark line from the shave head. Their appearance does not change if the skin is moved, thereby helping to identify them.*○*Implantation to the left is also possible by reversing the measurement.*
Using anatomical markers as a guide with a microsurgical drill, a 0.7 mm burr hole is created right in the middle of the skin incision (Figure 1 and Figure 2F,J,N).
○*Special care should be taken not to allow the steel burrs to penetrate into the mouse brain.*○*Blood vessel on the cranial surface should be avoided, particularly for intra-cerebellar injections.*○*A small amount of bleeding (20–50 µL) can be controlled by applying tissue adhesive glue to approximate the skin incision.*○*Bleeding was not commonly observed with intra-cerebral injection; less than 50 µL of blood loss can be seen with intra-cerebellar injection.*
Using a Hamilton gastight 10 µL syringe with a plastic stopper (1 mm in diameter) fixed at 3 mm from the tip (for intra-cerebral and intra-cerebellar tumors) and 5.2 mm (for intra-brain stem implantation) (Figure 2H,L), the position of the stopper is measured and ensured by a digital sliding caliber. Then, 2 µL of cell suspension (1 × 105 cells) for intra-cerebral and intra-cerebellar implantation and 1 µL (5 × 104 cells) for intra-brain stem implantation, which has been kept on ice, should be injected perpendicularly to the surface of cranial bone through the burr hole to a depth of 3 mm for both intra-cerebrum and intra-cerebellum tumors (Figure 1 and Figure 2) and 5.2 mm for brain stem engraftment. The cells are injected slowly, with a slight pause of 1–2 s prior to removing the needle to avoid back flow.Incision is closed with tissue adhesive glue.○*Since the skin incision is <1 mm, this step can be omitted.*
An ear tag with unique number is applied for future mouse identificationAfter surgery, all the animals are kept warm, clean, and dry throughout the immediate post-operative period.○*We normally give subcutaneous injection of buprenorphine SR (1 mg/kg), which is sustained release formulation of buprenorphine, to avoid causing stress to animals through repeated post-operation administration of pain killer.*○*Otherwise, pain killer should be administered follow institutionally approved animal protocols.*
Mice are observed until fully recovered from anesthesia before being returned to housing.
**D.** **Follow up:** the time and frequency of monitoring should follow an IACUC-approved protocol.
Post-operative monitoring: animals are monitored every day for the first 3 days to examine and document wound healing and their overall recovery on a surgery card.Long-term monitoring: post-tumor implantation, mice are checked at least three times a week for signs of neurological deficits (paralysis, uncontrolled rolling) or sickness (immobility, huddled posture, inability to eat, ruffled fur, self-mutilation, vocalization, dehisced wound, hypothermia, and/or weight loss). Mice showing any signs of these are then examined daily and euthanized following the approved animal protocol.Signs of tumor formation: tumor formation is typically indicated by the following clinical changes: moribund behavior (weight loss, lethargy, or decreased oral intake), enlarged head size, paralysis (less frequent), hunched posture or any other signs of illness. Mice displaying these signs should be sacrificed according to institutional procedures and their brains harvested and evaluated for the presence of tumors.Longitudinal monitoring of tumor growth: small animal MRI can be applied to monitor orthotopic xenograft growth. Due to the invasive nature of many malignant brain tumors, contrast enhancement is frequently required. The feasibility of transferring SCID mice to and from small animal imaging facility, the time needed for each scan, and the total cost should all be taken into consideration when planning for routine MRI scanning.
**E.** 
**Troubleshooting**



Tumor formation: If there is no clinical evidence of tumor formation in the mice within 4 to 6 months, imaging with MRI and CT would assist in assessing tumor growth. If small animal imaging is not available, and no signs of sickness are observed after 15 months, euthanasia of the cohort for pathological analysis should be considered.

Variations in tumor growth: since this is a freehand protocol, it is possible to induce some variations in tumor implantation. Using a needle with fixed depth and frequent checking of the stopper position is important for ensuring the same depth of the inoculated tumor cells. More practice can also reduce changes in the burr hole location, which will improve the reproducibility of the model. Maintaining consistent cell concentrations and implanting small volumes of tumor cells also help to reduce variation between different passages or among different models.
**F.** **Length of time for tumor formation**

Following this procedure, we have successfully developed 185 xenograft models from 407 pediatric CNS surgical samples, with a success rate of 72.3% in GBM, 64.2% in medulloblastoma, 50% in ATRT, 33.8% in ependymoma, and 11.6% in low-grade gliomas (Figure 1) [39]. DNA genotyping and gene expression profiling revealed that the xenograft tumors were genetically similar to the original patient tumors, and, furthermore, serial in vivo sub-transplantation did not cause a significant change in the genomic profile of the tumors [33,45,47]. Time to tumor formation after the first passage may vary depending on tumor histology. Mice were implanted with 1 × 10^5^ cells, the earliest tumor formation, i.e., the survival time of the first mouse that became moribund due to tumor growth and had to be euthanized from 38 to 175 days (median 74 days) in 10 ATRT models, from 55 to 106 (median 88) days in 4 embryonal tumors with multilayer rosettes (ETMR), from 34 to 233 (median 121) days in 29 GBM, and from 43 to 320 (median 121) days in 44 MB models, and from 56 to 304 (median 189) days in 26 ependymoma models (Figure 1C). In addition to inter-tumoral differences, mice implanted with the same tumor cells may exhibit different tumor formation rates. The survival times of the last tumor-bearing mouse of each model were as long as 290 days in ATRT, 379 days in ETMR, 463 days in both GBM and MB, and 490 days in EPN. Therefore, we should exercise patience and carefully monitor SCID mice.**G.** **Serial sub-transplantations in vivo in mouse brains**

A large cohort of animals from a well-characterized and clinically relevant xenograft model is indispensable for both biological studies and pre-clinical drug screening. The limited availability of tumor tissues from pediatric patients only allows direct transplantation into small numbers of animals. It is therefore important to perform sub-transplantations from established xenograft tumors to expand xenograft availability and numbers of animals. Tumor cell yields from different types of pediatric tumors, however, are different. A PDOX tumor of medulloblastoma and GBM can generally yield 5–20 million viable cells, ependymoma 2–10 million, and ATRT 2–10 million cells. The overall tumor cell yields from DIPG models are lower, usually in the range of 1–2 million cells.


**
Procedure
**
To harvest mouse brain, a tumor-bearing mouse is put under deep anesthesia through i.p. injection of Beuthanasia-D special (100 µL/20–15 g).Whole brains of donor mice are aseptically removed, coronally cut into halves, and transferred back to the tissue culture laboratory in cold (4 °C) growth medium (DMEM + 10% FBS and antibiotics).Xenograft tumors are then dissected from mouse brains.○*Since medulloblastoma and glioblastoma tissues are soft and fragile, tumor cores can often break up by gentle tapping with the back of a scalpel.*○*Intra-brain stem DIPGs may not be easily visible even after removing the cerebellum.*
Tumors are mechanically dissociated into single cell suspensions and injected into the brains of recipient SCID mice as soon as possible (within 60 min of tumor removal), as described above.○*To facilitate the measurement and monitoring of changes in tumor growth rate, it is recommended to implant a fixed number of viable tumor cells. We usually use 1 × 105/mouse.*○*Dead cells or debris do not need to be cleaned.*




**
Results of tumor growth
**


The growth rate of xenograft tumors tends to increase in the second passage and become stabilized thereafter [33,35,36]. The mean survival times are reduced to 58–109 days in medulloblastoma and to 60–129 days in GBM models. The animal survival times also tend to cluster more tightly, and the animals should be monitored more frequently (at least one to two times a day) for signs of sickness or neurological deficit. Due to the malignant and progressive nature of brain tumors, the rapid deterioration of animals’ health condition or sudden death (overnight, or even several hours later) can happen, particularly in models of DIPG and GBM.
**H.** **Long-term cryopreservation of xenograft cells in liquid nitrogen**

Preserving xenograft tumor cells by cryopreservation in liquid nitrogen is central to the ability to provide a sustainable supply of xenograft models whenever needed and to minimize the genetic drift and phenotypic changes induced by repeated serial passaging in vivo.


**
Supplies
**

**Ingredient**

**Vender**

**Cat. No**

**Unit Size**
DMEMVWR45000-3046 × 500 mLFetal bovine serum (FBS); premium, heat-inactivatedAtlanta BiologicalsS11150H500 mLDimethyl sulphoxide (DMSO) SigmaD2650100 mL



**
Procedures for cryopreservation
**
Freshly harvested xenograft tumors are mechanically dissociated to prepare cell suspensions.Single cells are collected by passing cell suspensions through 40 and 100 µ cell strainers.Cells are counted with trypan blue.Pellet cells through centrifuge at 1000 rpm for 5 min.Cell pellets are re-suspended with freshly prepared cell-freezing medium (DMEM medium supplemented with 10% fetal bovine serum and 10% dimethyl sulfoxide) at 1–3 × 10^6^ cells/mL.A 1.5 mL cell suspension is aliquoted into cryovials and stored at −80 °C overnight.Cryovials are transferred into liquid nitrogen.The tumor cell information (*model ID, mouse number, number of viable cells, date of harvest*) and location (*liquid nitrogen tank number, rack number, box number and location*) are then documented.



**
Procedure for cell retrieval
**
Pre-warm a water bath to 37 °C.Remove the cryovial of cells from liquid nitrogen wearing proper personal protections.Immediately put the vial in the water bath (37 °C) and swirl with your hand in the water bath until there is no ice left in the tube.Clean the outside of the vial with 70% ethanol before placing it in a tissue culture hood.Transfer the cell suspension from the cryovials to a T75 flask, and add 25 mL fresh medium (either FBS-based or CSC medium) slowly (drop-wise) while mixing.
-*This step is important for avoiding sudden changes of osmolality, which can significantly reduce the viability of primary xenograft cells. The slow adding of fresh medium should take ~2 min.*-*Because the primary xenograft cells are still very fragile at this stage, we normally do not wash the cells to remove DMSO.*
Pipette the cells up and down several times, and estimate cell viability through trypan blue staining.
**I.** 
**Characterization of PDOX Tumors**



Detailed characterization of xenograft tumors is needed to confirm that the PDOX tumors originated from the human tumors, replicate the key histopathological phenotypes, and maintain the key genetic abnormalities of the original patient tumors. Since PDOX tumors are implanted in mouse brains, additional assays are also needed not only to quantify the relative abundance but also to purify human tumor cells and/or the host mouse brain cells for downstream molecular analysis.


**1. Histopathological Examination**


The following protocols have been developed and optimized by our laboratory for hematoxylin–eosin (HE)/immunohistochemistry (IHC) experiments using paraffin-embedded tissue samples.


**1a. Paraffin embedding of whole mouse brain**
(a)Tissue processing/fixation
The whole mouse brain is harvested and placed into a vial of fixative containing 10 mL zinc formalin (Cat. #.5701ZF, Thermo Scientific, Waltham, MA, USA) for 24 h at 4 °C with occasional inversion.It is important that brains are handled carefully, remain intact, and are put into fixative as soon as possible when harvested.We have seen better preservation of tumor cell antigenicity since switching from regular formalin to zinc formalin.
After 24 h, tissue is transferred into a cassette (mega cassette), marked with #2 pencil on three sides, immersed in 75% ethanol, and stored at 4 °C at least overnight. Mouse brains can be kept in 75% ethanol for several months.
(b)Dehydration: This is achieved through a step-wise increase in ethanol concentration at room temperature.
Step 1: 95% Ethanol: 2 h;Step 2: 95% Ethanol: 2 h;Step 3: 100% Ethanol: 4 h.
(c)Clearing:
The tissue cassette is transferred to a glass jar filled with chloroform in a chemical hood.The jar is covered with aluminum foil and kept overnight.
(d)Infiltration:
Next morning, tissue is transferred to pre-melted Paraffin I/III (Thermo Scientific) at 65 °C for 2 h.Infiltration is repeated in a second round of fresh Paraffin I/II for an additional 2 h.Embedding is completed using a Leica EG1160 embedding center, dispenser, and hot plate (Leica Biosystems).A stainless steel mold is sprayed with a base mold release agent (Richard-Allan Scientific/Thermo Fisher Scientific, Waltham, MA, USA).The mold is filled with melted paraffin on the hot plate.The tissue is transferred with hot forceps and oriented upside down in the center of the mold.The white plastic form is placed on top of the mold and filled with melted paraffin.the plastic form is removed from the heat.The plastic form is held gently for a few seconds to allow for the formation of a scum of paraffin to “fix” the plastic form in place.The mold is transferred onto a refrigerated surface and allowed to sit for at least 2 h to allow the paraffin to completely solidify.the tissue block is removed from the mold with its attached cassette.The tissue block is then ready for sectioning (at 5 µm thickness using a microtome).




**1b. H&E staining procedure**
Incubate paraffin slides at 60 °C × 15 min.Dewax by immersing the slides in
Xylene for 2 min;Xylene for 2 min;Xylene for 2 min.
Rehydration
100% alcohol for 2 min;95% alcohol for 2 min;70% alcohol for 2 min;50% alcohol for 2 min;DI water for 2 min.
Harris hematoxylin for 5 min;Wash in running water for 2 min;Bluing agent for 30 s;DI water for 2 min;95% alcohol for 2 min;Eosin Y for 15 s (if using 0.2%, stain for 70 s);95% alcohol for 2 min;100% alcohol for 2 min;Xylene for 2 min;Xylene for 2 min;Mount media on a cover slip.



**1c. Immunohistochemical (IHC) staining**



**1c.1. Confirmation of human origin of PDOX tumors**


To confirm the human origin of xenograft tumors in mice, IHC is applied on paraffin-embedded whole brain tissue sections using human-specific antibodies. Positive staining is indicative of the existence of implanted human tumors.
Antibodies:
○Human mitochondria (MT) (EMD Millipore. MA, 1:200 dilution);○Human vimentin (VIM) (Clone V9, DAKO, CA, 1:200 dilution);○Human Ki-67 (Abcam, MA, 1:100 dilution).
IHC: This is achieved through a standard protocol, as described previously, using a Vectastain Elite kit (Vector Laboratories, Burlingame, CA, USA). After slides are incubated with primary antibodies for 90 min at room temperature, the appropriate biotinylated secondary antibodies (1:200) are applied and incubated for 30 min, and the final signal is developed using the 3,3′-diaminobenzidine (DAB) substrate kit for peroxidase. The IHC staining is assessed by combining the intensity and extent of immunopositivity (Figure 3).
○MT positivity:
▪Strong and homogeneous staining is frequently seen in most medulloblastoma and ependymoma models as well as a radiation-induced anaplastic astrocytoma [33,35] (Figure 3A);▪Tumor cells in GBM and DIPG models usually exhibit different levels of MT positivity, particularly in the tumor core area (Figure 3A).
○Vimentin positivity:
▪Strong positive reactions are seen in most of the medulloblastoma, ependymoma, PXA, and GBM models [36,49], whereas in DIPG models, vimentin-negative cells are seen (Figure 3B);▪Invasive tumor cells from GBM, DIPG, and ependymomas are frequently stained strongly (+++) positively, even single invasive tumor cells [49] (Figure 3B).



**2. Quantitation and purification of human PDOX cells**


**2a. Quantitative PCR (qPCR) estimation of mouse cell contamination**: Due to the invasive nature of malignant brain tumors, mouse cells’ contamination in dissociated PDOX xenograft cells occurs universally. To determine the percentage of mouse contamination, a quantitative PCR-based analysis is used.
Genomic DNA extraction: This can be achieved with any standard protocol. In our lab, DNA is extracted from harvested cells/tissues using a Promega Wizard genomic DNA extraction kit (Promega, WI), Trizol reagent (Thermo Fisher Scientific, MA) or an Allprep DNA/RNA mini Kit (Cat. NO. 80204, QIAGEN) following the manufacturer’s instructions. Quantification of DNA concentrations is performed using a NanoDrop-1000 (Thermo Fisher Scientific).Quantitative PCR analysis: Human and mouse-specific primer pairs for the target prostaglandin E receptor 2 (PTGER2) are used [50]. Primer sequences include human-specific forward (hPTGER2-F): gctgcttctcattgtctcgg, mouse-specific forward (mPTGER2-F): cctgctgcttatcgtggctg, and common reverse (hmPTGER-R): gccaggagaatgaggtggtc. The PCR product sizes are 189 bp for humans and 186 bp for mice. Reaction mix contained SYBR Green select master mix (ABI, #4472908), respective primers, and 10 ng of total genomic DNA in a total reaction volume of 10 μL. qPCR was carried out using a StepOnePlus Real-Time PCR System (Applied Biosystems, #4376600) following a standard protocol, i.e., 50 °C × 2 min, 95 °C × 10 min, 40 cycles of (95 °C-15 sec, 61 °C-1 min). Agarose gel and a melting curve are used to check product size and primer specificity.Data analysis and statistics: percentages of mouse and human gDNA content are calculated as follows: 1. CT_hPTGER2_ − CT _mPTGER2_ = ΔCT_human_ and CT_mPTGER2_ − CT _hPTGER2_ = ΔCT_mouse_; 2. (ΔCT_human_/(ΔCT_mouse_-1))*100. This results in an estimation in percent of the human and mouse gDNA in each sample. In cases in which no human or mouse gDNA is detected, the value is set at 100%. Each sample is analyzed at least in triplicate, and mean CT values as well as normalized gDNA content are shown (Figure 4A,B).

**2b. FACS purification of human xenograft cells**: Mouse cell contamination in tumor cells dissociated from mice xenografts occurred universally in our PDOX models. To determine the percentage of mouse contamination and further eliminate these mouse cells, we utilized a cocktail of human-specific and mouse-specific antibodies against cell surface markers to separate human xenograft cells from mouse brain cells [49].
Human-specific antibodies: fluorochrome-conjugated antibodies against human HLA-ABC and HLA-DR (Miltenyi Biotec, Cologne, Germany).Mouse-specific antibodies: CD19, CD133, CD40, and CD140 (Biolegend, San Diego, CA, USA) are employed to analyze the percentage of human or mouse cells contained in samples.Decontamination of mouse cells: Based on the above results, FACS is used to sort out HLA-positive/mouse negative and HLA/mouse double-negative cells for future uses (Figure 4C).
○*Limitations: Not all brain tumors express HLA-ABC, -DR.*



**3. Molecular characterization:**
Global profiling: To compare xenograft tumors with patient samples at the molecular level, whole-genome sequencing, single-cell sequencing, DNA methylation profiling, gene copy number, and gene expression profiling (RNAseq) can be performed during serial transplantations and compared with the original patient tumors (Figure 5) [33,45,46,47,49,51,52]. Since mouse cell contamination is always a possibility, procedures or protocols should be considered to achieve pre-omics analysis decontamination (i.e., the purification of human xenograft cells before extraction of DNA and/or RNA) or post-omics filtration (i.e., digital filtering of mouse cell sequences).PDOX model authentication: Similar to human cell lines, PDOX models should also be regularly authenticated [53]. Short tandem repeat (STR) DNA profiling is a reliable approach that can be repeatedly applied to PDOX model management [43,45].



**4. Initiation of cell cultures**


PDOX tumor cells dissociated from mice xenografts can serve as a sustained source of cells (in addition to primary patient tumor cells) to establish in vitro tumor cell lines successfully. Tumor cells can be cultured in different culture media with penicillin/streptomycin at 5% CO_2_ incubator at 37 degrees. In addition to classic monolayer cultures, 3D growth of neurospheres (which are enriched with putative cancer stem cells) and tumor organoids can be initiated as well.

a. Monolayer cells: DMEM media supplemented with 10% fetal bovine serum;

b. Serum-free neurosphere culture medium [54,55,56,57,58]:DMEM/F12: 500 mL (VWR, 45000-344)rhEGF: add 25 µg to 500 mL (final concentration: 50 ng/mL) (R&D Systems, 236-EG)rhFGF-basic: add 25 µg to 500 mL (final concentration: 50 ng/mL) (R&D Systems, 4114-TC-01M)B27 supplement: use a whole vial (final concentration: 1x) (Invitrogen, 17504-044)N2 supplement: use a whole vial (final concentration: 1x) (Invitrogen, 17504-048)Penicillin–streptomycin: 10 mL (Invitrogen, 15140-122)L-Glutamine-200 mM: 5 mL (original 100x) (optional) (Invitrogen, 25030-149)*Due to the invasive growth of PDOX tumors in mouse brains, mouse cells are frequently present in the dissociated xenograft tumor cells. Therefore, determination of the cell origin (mouse or human) should be carried out periodically either with RT-PCR using human-specific and mouse specific-genes, or FCM, or both.*


**5. Statistical Considerations for Pre-Clinical Drug Testing**


For pre-clinical evaluation of single-agent treatment, sample size calculations for determining treatment efficacy must contain adequate numbers in order to detect difference and account for unanticipated complications (infection, surgical or anesthesia complications). Response to therapy may vary as tumors progress and potentially metastasize. Therefore, most often, experimental design includes two cohorts, one with small tumors that have been implanted for 2 weeks (1–2 mm in size at time of treatment) and one with large tumors (4–6 mm) that are allowed to grow for 4 weeks prior to the initiation of treatment. For traditional in vivo drug testing, sample size estimation is completed with SigmaPlot 14. For example, to detect a *difference in means* > 0.10, *standard deviation* of 0.06, *power of detection* of 0.90, and an *alpha* (the risk of a false positive conclusion) of 0.050, we usually need 10 mice per group [37,40,48,52]. Each cohort includes a control group (n = 10), treated with vehicle only; an experimental group (n = 10), and a biology group. The biology group will include three to five mice for each desired time point of evaluation.

## 4. Conclusions

**Key points:** (1) This technique can be broadly applied to all types of brain tumors. (2) The protocol is easy to follow in order to generate large cohorts of tumor-bearing mice. (3) The protocol meets the needs of biological studies and preclinical drug testing.

**Characteristics of the study:** (1) *Rapid:* implantation of one mouse takes <60 s. A team of three investigators can implant 150–200 mice in 4 h; (2) *Safe:* the surgically related death rate is less than 0.5% after ~40,000 orthotopic implantations; (3) *Precise depth of tumor cell engraftment and highly reproducible*: said depth is determined by a stopper. (4) *Only a small amount (as few as 0.1 million) patient tumor cells are needed*. (5) *High tumor take rate* (greater than 60–70%); (6) *No need for extra equipment* (as ours is a freehand procedure); (7) *Biologically accurate*: brain tumors can be grown in corresponding locations in the mouse brain; (8) *Clinically relevant*: the existence of the blood–brain barrier enables evaluation of the efficacy of in vivo drug delivery and overall therapeutic efficacy; (9) *Remarkably increased success in developing matching in vivo models*: PDOX tumors provide a resourceful platform for establishing matching panels of in vitro models.

## Figures and Tables

**Figure 1 cancers-16-01716-f001:**
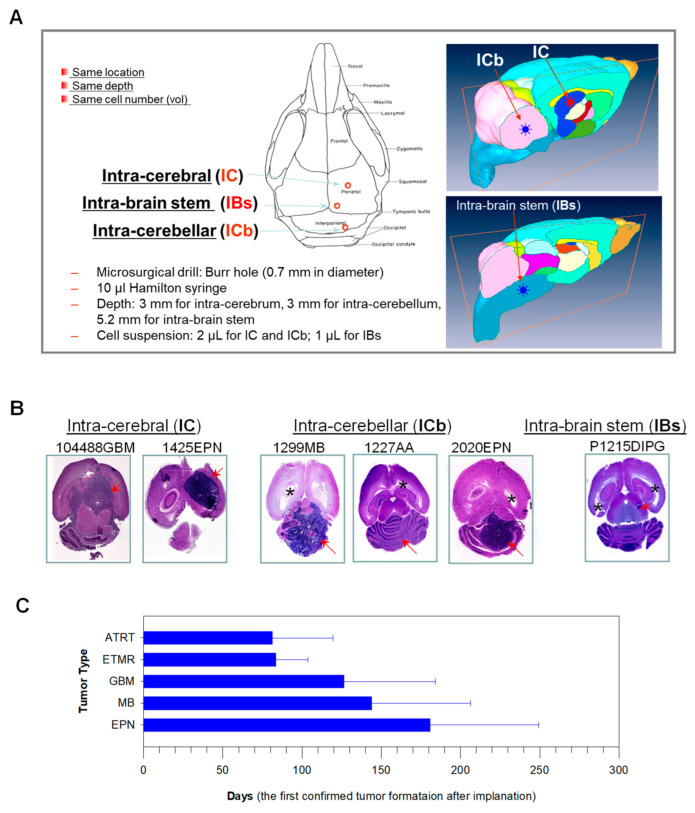
Orthotopic implantation of brain tumor cells into matched locations in mouse brains. (**A**). Outline of surgical procedures. For supratentorial tumors such as GBM, the burr hole (*blue circle*) is made on the right parietal bone 1 mm to the midline and 2 mm anterior to the bregma occipital line for intra-cerebral (***IC***) implantation. For infra-tentorial tumors such as medulloblastoma, the burr hole (*red circle*) is made on the right inter-parietal bone 1 mm to the midline and 1 mm posterior to the bregma occipital line for intra-cerebellar (***ICb***). For brain stem tumors, the burr hole is made at the right corner of the midline and bregma occipital line for intra-brain stem (IBs) injection. Tumor cells will be injected at a depth of 3 mm below the outer surface of the skull for IC (near the right caudate nucleus) and ICb (in the middle of the right hemisphere of the cerebellum) and 5.2 mm for IBs (in the middle of pons) implantation. The drawing of the mouse skull was adopted from Mouse Genome Informatics (http://www.informatics.jax.org/cookbook/chapters/skeleton.shtml, accessed on 1 December 2012). (**B**). Formation of IC, ICb, and IBs xenografts of glioblastoma (GBM), ependymoma (EPN), medulloblastoma (MB), anaplastic astrocytoma (AA), and diffuse intrinsic pontine glioma (DIPG) as detected with H&E staining on paraffin sections of whole mouse brains. * Enlarged ventricle indicative of hydrocephalus. (**C**). Summary of the times needed for the earliest tumor formation, i.e., the survival time of the first mouse that became moribund due to tumor growth of five major types of pediatric brain tumors.

**Figure 2 cancers-16-01716-f002:**
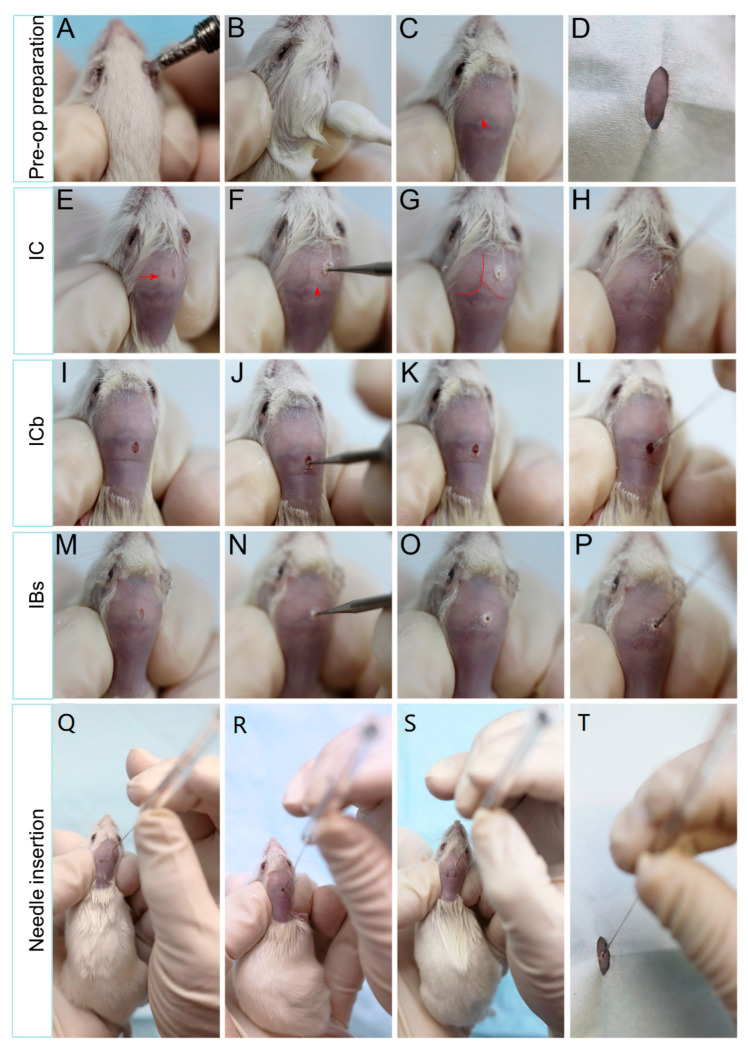
Representative images showing the freehand tumor implantation into mouse brains. Pre-operative preparation includes the application of eye ointment (**A**), hair remover (**B**,**C**), and a sterile sheet (**D**). Skin incision, burr hole drilling, and needle insertion of IC (**E**–**H**), ICb (**I**–**L**), and IBs (**M**–**P**) are shown. Note the mid-line (arrow) (**E**) and bregma occipital line (arrowhead) (**F**) that are visible from the shaved skin as highlighted in G. Needles containing tumor cells are inserted perpendicular to the skull surface (**Q**–**T**).

**Figure 3 cancers-16-01716-f003:**
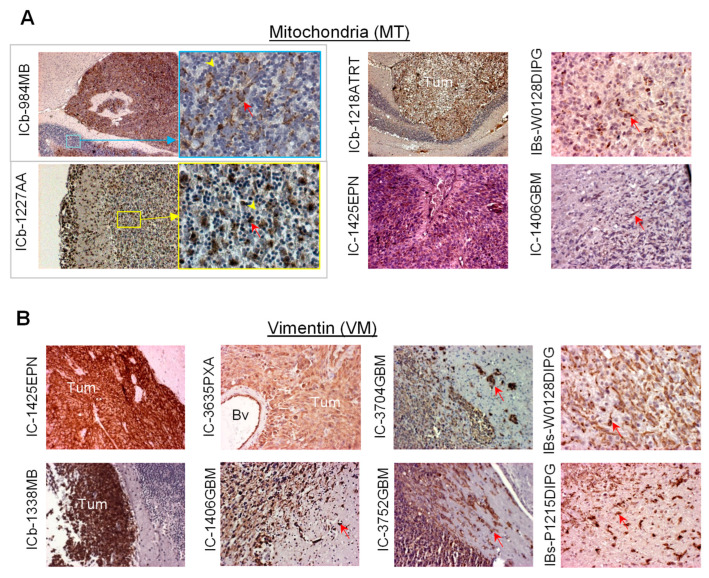
Representative images of IHC staining of PDOX tumors with human-specific antibodies against mitochondria (**A**) and vimentin (**B**). Tumors (*Tum*), positively stained tumor cells (*arrow*), and blood vessels (*Bv*) are indicated. Note that pediatric GBM is replace by high-grade glioma (of different subtypes) in WHO CNS Tumor Classification (2021). Bar = 100 µM.

**Figure 4 cancers-16-01716-f004:**
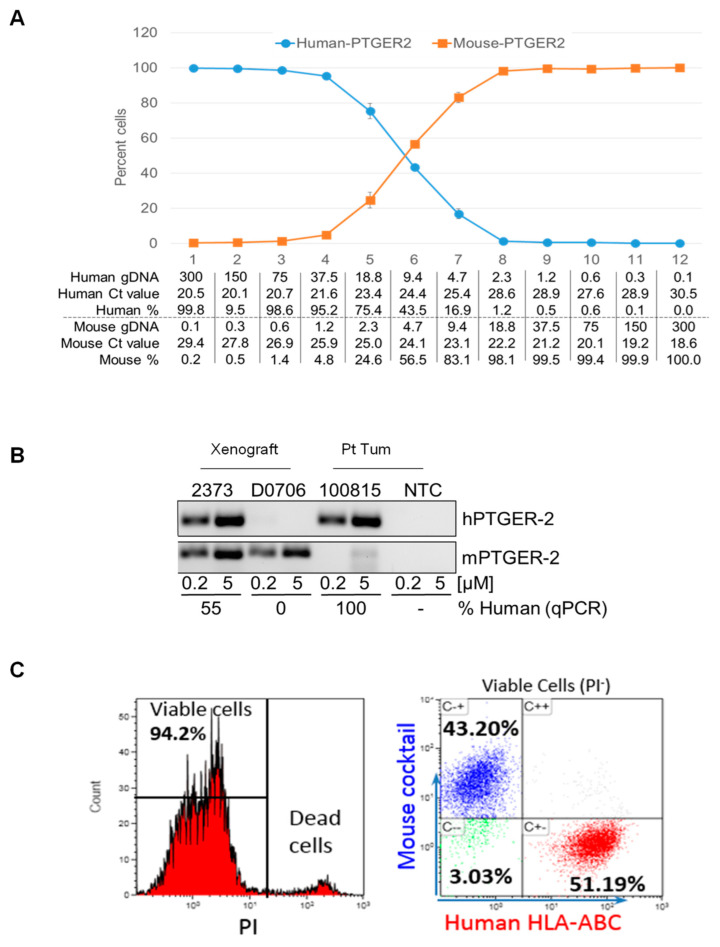
Analysis of human and mouse cell abundance in PDOX models. (**A**) Characterization of qPCR am-plification efficiency by mixing human (*blue line*) and mouse (*orange line*) DNA through serial dilu-tions. (**B**). Visualization of PCR and qPCR products on agarose gel in xenografts with (***2373***) and without (***D0706***) human DNA using a human tumor (***100815***) as a positive control and no tumor cell (***NTC***) as a negative control. (**C**) Flow cytometric analysis of human and mouse cells using hu-man-specific antibodies against HLA-ABC and an antibody cocktail (CD19, CD133, CD40 and CD140) specific to mouse brain cells.

**Figure 5 cancers-16-01716-f005:**
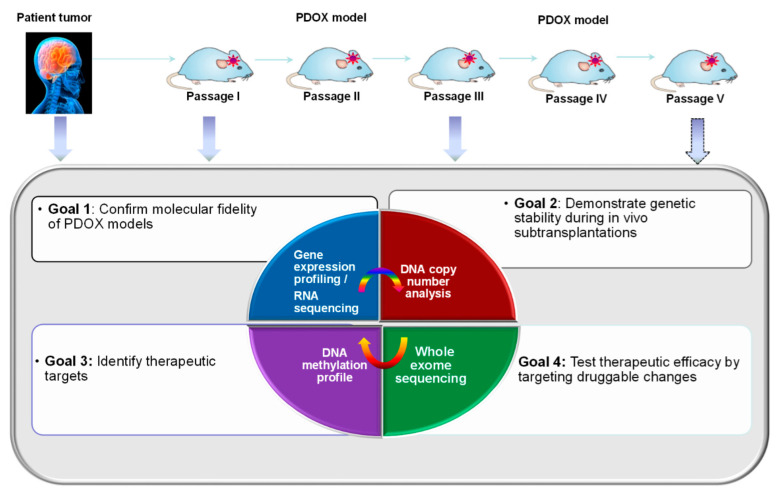
Strategies for comprehensive molecular characterization of PDOX models during serial in vivo sub-transplantations.

**Table 1 cancers-16-01716-t001:** PDOX models of brain tumors.

	Clinical Stage			
Tumor Type	At Diagnosis	Relapse	Autopsy	Sub Total
** *Pediatric* **				
Medulloblastoma	54	3	5	62
High grade glioma	18	7	10	35
DIPG	4		14	18
Ependymoma	13	10	1	24
ATRT	11	1	2	14
ETMR	3		1	4
CNS EFT-CIC	1	2		3
CNS-Germinoma		2		2
PXA	1		1	2
Ganglioglioma	1			1
Pinealblastoma	1			1
** *Adult* **				
GBM (adult)	21			21
Meningioma (adult)	2			2
*Total*	130	24	34	**188**

DIPG = Diffuse intrinsic pontine glioma; ATRT = Atypical teratoid/rhabdoid tumor; ETMR = embryonal tumors with multilayered rosettes; CNS EFT-CIC = Ewing sarcoma family with CIC alteration; PXA = pleomorphic xanthosarcoma; GBM = Glioblastoma.

## Data Availability

All data generated or analyzed during this study are included in this manuscript.

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
