# Peer review of "Direct Implantation of Patient Brain Tumor Cells into Matching Locations in Mouse Brains for Patient-Derived Orthotopic Xenograft Model Development"

_cancers, 2024, doi:10.3390/cancers16091716_

Round 1
Reviewer 1 Report (Previous Reviewer 2)
Comments and Suggestions for Authors
The manuscript provides a detailed protocol for the establishment and characterization of patient-derived orthotopic xenograft (PDOX) models of pediatric central nervous system (CNS) tumors. These models are crucial for preclinical drug screening and studying tumor biology. The protocol covers various aspects, including tumor implantation and growth monitoring, length of time for tumor formation, cryopreservation and characterization of PDOX tumors, and initiation of cell culture. Lastly, the authors briefly discussed the sample size estimation in preclinical drug testing. Overall, the manuscript offers a comprehensive guide for developing and characterizing PDOX models of pediatric CNS tumors. To provide a more comprehensive understanding of PDOX models and their utility in preclinical cancer research, the authors could conduct the following experiments:
1. Besides immunohistochemical (IHC) staining, the authors could conduct experiments to validate the genetic similarity between PDOX tumors and original patient tumors through additional molecular analyses like whole-genome sequencing, single-cell sequencing?
2. The authors could perform some functional studies to evaluate their established PDOX models, including drug efficacy of novel treatment modalities or standard-of-care therapies, gene knockdown or overexpression, pathway inhibition, or immune checkpoint inhibition.
3. Have authors assess the metastatic potential of PDOX tumors by monitoring the development of distant metastases in relevant organs?
Minor:
The subheadings in the manuscript require restructuring. Duplicate titles are present, such as "Materials" appearing at both Line 160 and Line 417. Additionally, inconsistent labeling methods are used, including A, B, C, as well as 3.1, 3.2, 1. 2. This inconsistency contributes to the overall disorganization of the manuscript.
Author Response
The manuscript provides a detailed protocol for the establishment and characterization of patient-derived orthotopic xenograft (PDOX) models of pediatric central nervous system (CNS) tumors. These models are crucial for preclinical drug screening and studying tumor biology. The protocol covers various aspects, including tumor implantation and growth monitoring, length of time for tumor formation, cryopreservation and characterization of PDOX tumors, and initiation of cell culture. Lastly, the authors briefly discussed the sample size estimation in preclinical drug testing. Overall, the manuscript offers a comprehensive guide for developing and characterizing PDOX models of pediatric CNS tumors. To provide a more comprehensive understanding of PDOX models and their utility in preclinical cancer research, the authors could conduct the following experiments:
Besides immunohistochemical (IHC) staining, the authors could conduct experiments to validate the genetic similarity between PDOX tumors and original patient tumors through additional molecular analyses like whole-genome sequencing, single-cell sequencing?
Response: Yes, this is exactly what we should do and have done as we highlighted in Figure 5. Some data have been published before.
The authors could perform some functional studies to evaluate their established PDOX models, including drug efficacy of novel treatment modalities or standard-of-care therapies, gene knockdown or overexpression, pathway inhibition, or immune checkpoint inhibition.
Response: Yes, indeed! This is the objective of all our efforts in developing PDOX models. Many drugs have been tested and some functional analysis of genes and pathways completed. But we have not been able to examine the efficacy of immune therapy due to the lack of immunity (no T, no B and no NK cells) in our SCID mice.
Have authors assess the metastatic potential of PDOX tumors by monitoring the development of distant metastases in relevant organs?
Response: We have checked the CSF spread in some medulloblastoma models (including spread into the spinal cord), but have not examined the development of distant metastasis as it is very common for brain tumor. It may worth a quick look in the future.
Minor:
The subheadings in the manuscript require restructuring. Duplicate titles are present, such as "Materials" appearing at both Line 160 and Line 417. Additionally, inconsistent labeling methods are used, including A, B, C, as well as 3.1, 3.2, 1. 2. This inconsistency contributes to the overall disorganization of the manuscript.
Response: Yes, this is a very good suggestion. We changed the “Materials” to “Supplies” in Line 417. We have re-structured and re-ordered the subheadings to make the labelling consistent.
Reviewer 2 Report (New Reviewer)
Comments and Suggestions for Authors
Major points:
(Reviewer expertise in Glioblastoma)
The current study entitled “Direct implementation of patient brain tumor cells into matching location in mouse brains for patient-derived orthotopic xenograft model development” by Qi Lin et al. is aimed to make easy protocol to generate mouse model without requirement of stereotactic frame device and proposed clinically relevant and biologically accurate (CRBA) mouse model for brain tumors to understand preclinical settings.
However, I have some remarks which need to be addressed by the authors about developing these brain tumor model:
There is lack of information about the brain regions where the samples were collected, the authors should ensure the heterogeneity are not from the brain region differences, the compassion is only reasonable when the samples are from the same/similar brain regions.
Authors have claimed the success ratio for GBM is >70%, however, glioblastoma is itself hard to find the exact location of point of origin. I wonder, if authors have all the clinical information at the time of surgery and exact tumor location and type of tumor in all the mentioned model. Explanation required.
“Precise depth of tumor cell engraftment: the depth is determined by a stopper attached to a Hamilton gas tight needle at pre-determined depth.” Authors need explain the instrument “stopper” with how precise depth of the measurement for inserted needle. Does this change with mouse body weight?
In figure 1: Authors should mention the tumor formation rate, tumor volume, and survival curve of these brain tumor mouse model.
In Figure 1 B, image 2020EPN is the same image from their previous work (https://doi.org/10.1038/s41467-022-34514-z). (Ref?)
In figure 1 legend C) “Summary of the times needed for the earliest tumor formation, i.e., the survival time of the first mouse that became moribund due to tumor growth of five major types of pediatric brain tumors.”. Do authors mean this by detection of tumor by MRI or overall neurological symptoms?
Line 361-364- “DNA genotyping and gene expression profiling revealed that the xenograft tumors were genetically similar to the original patient tumors, and, furthermore, serial in vivo sub-transplantation did not cause a significant change in the genomic profile of the tumors” Do authors have information if RNA-seq showing maintenance of similar transcriptomic signatures across all the brain tumor model to recapitulate the exact genetic alterations in patients and PDX models?
In figure 3, does authors use the same mouse tag number (IC-1406GBM) from their previously published article (Ref-49)?
In Figure 3, Ki-67 panel is missing.
Missing scale bar in Figure 3.
In figure 4 C, Viable cell (PI-), second panel annotation is confusing (Viable cell PI-)
Line 647 (Heading)- Typo (dRug)
Line 648-651- If possible, authors can simplify this statement.
Author’s self-citation looks self-aggrandizement. I may suggest them to look for similar articles from other research groups as well to be propose more generalized protocol for the benefit of readers.
Author Response
The current study entitled “Direct implementation of patient brain tumor cells into matching location in mouse brains for patient-derived orthotopic xenograft model development” by Qi Lin et al. is aimed to make easy protocol to generate mouse model without requirement of stereotactic frame device and proposed clinically relevant and biologically accurate (CRBA) mouse model for brain tumors to understand preclinical settings.
However, I have some remarks which need to be addressed by the authors about developing these brain tumor model:
There is lack of information about the brain regions where the samples were collected, the authors should ensure the heterogeneity are not from the brain region differences, the compassion is only reasonable when the samples are from the same/similar brain regions.
Response: Getting multiple tumor tissues from different and very-well defend regions would be the dream of any brain tumor investigator, let alone the model developers. We totally agree with the reviewer this is extremely important. However, we have to admit that it is equally (and extremely) difficult to achieve this goal. In fact, we have not been able to collect a single tumor with such regional information. All we had wished and is still wishing now is to have a small piece of tumor tissue (usually 1-2 mm in diameter) for the initiation of tumor implantation. Our neurosurgeons and pathologists are extremely supportive, we will try to collect region information in the future.
Authors have claimed the success ratio for GBM is >70%, however, glioblastoma is itself hard to find the exact location of point of origin. I wonder, if authors have all the clinical information at the time of surgery and exact tumor location and type of tumor in all the mentioned model. Explanation required.
Response: This is a good point, and we agree that the “exact location of point of origin” is very difficult to define. It is because we recognize such difficulties and because we are not trying to replicate “tumor of orgin”, we set to implant brain tumors to their “matching locations”, which is referring to the cerebrum, cerebellum and brain stem. This is also the required tumor location information from our neurosurgeons and/or pathologist when the tumor samples were collected at the time of surgery. The following sentence was added in Line 244:
- Transplant: Brain tumor cells will be transplanted into the matching anatomical location of the original patient tumor in the mouse brains, i.e., cerebral tumors into mouse cerebra, cerebellar tumors into mouse cerebella, and brain stem tumors into mouse brain stem.
“Precise depth of tumor cell engraftment: the depth is determined by a stopper attached to a Hamilton gas tight needle at pre-determined depth.” Authors need explain the instrument “stopper” with how precise depth of the measurement for inserted needle. Does this change with mouse body weight?
Response: It is detailed in the method (line 289 on page 7) and shown in Fig 1 H and 1L. It is small plastic tube fixed to the needle.
Using Hamilton gastight 10-µL syringe with a plastic stopper (1 mm in diameter) fixed at 3 mm from the tip (for intra-cerebral and intra-cerebellar tumors) and 5.2 mm (for intra-brain stem implantation) (Fig. 1H, 1L). The position of the stopper is measured and ensured by a digital sliding calibre.
This depth remains the same for all the animals regardless of mouse body weight. To minimize the differences of the depth of tumor implantation due to body weight, we tried to standardize the age of animals (5-8 weeks for pediatric brain tumors and 10-12 weeks for adult brain tumors, Line 167). The following note was added in Line 170, page 4:
- Standardizing animal age can also reduce variabilities of animal body weight and head size. The latter is also important for minimizing the depth differences of tumor implantation.
In figure 1: Authors should mention the tumor formation rate, tumor volume, and survival curve of these brain tumor mouse model.
Response: Yes, these are critical questions about model characterization. For all our model development papers, we have included all of these and many other information. We are, however, running into some paradoxical situations, because we are not supposed to cite too many of our own paper.
In Figure 1 B, image 2020EPN is the same image from their previous work (https://doi.org/10.1038/s41467-022-34514-z). (Ref?)
Response: Yes, this is something we have tried very hard to avoid and we appreciate the reviewer for identifying this (thanks a lot). A new image is inserted from different ependymoma (ICb-1499EPN). And, the tumor ID of 2020EPN should be 2002EPN as we published in the Nat Comm. Once again, we thank the reviewer for catching this error for us.
In figure 1 legend C) “Summary of the times needed for the earliest tumor formation, i.e., the survival time of the first mouse that became moribund due to tumor growth of five major types of pediatric brain tumors.”. Do authors mean this by detection of tumor by MRI or overall neurological symptoms?
Response: We meant the “animal survival times” determined by overall health condition and neurological symptoms. By our approved animal protocol means the animals will be euthanized “when animal develop neurological deficits (symptoms or signs) or become moribund”. MRI is not routinely performed for tumor growth screening because it usually takes longer time (30-60 min) per mouse and frequently need i.v. injection of contrast.
Line 361-364- “DNA genotyping and gene expression profiling revealed that the xenograft tumors were genetically similar to the original patient tumors, and, furthermore, serial in vivo sub-transplantation did not cause a significant change in the genomic profile of the tumors” Do authors have information if RNA-seq showing maintenance of similar transcriptomic signatures across all the brain tumor model to recapitulate the exact genetic alterations in patients and PDX models?
Response: This is a great question, although it is more related to the model characterization. We have multiple publications showing the similarities of genetic and epigenetic features between PDOX models and the originating patient tumors. But we will not say it is the “exact genetic alteration”, given the intra-tumoral heterogeneity of the originating patient tumors and the adaptive, progressive and selective changes during the serial in vivo subtransplantations.
In figure 3, does authors use the same mouse tag number (IC-1406GBM) from their previously published article (Ref-49)?
Response: Again, this is a superb observation. The tumor model ID, 1406, is the same but mouse number (mouse tag?) is different. Every animal is given a unique mouse number in our database. This tumor was originally diagnosed as GBM. Since the WHO CNS Tumor Classification (2021) relaced GBM with HGG, we thought it make more sense to adjust the diagnosis to HGG in Table1. We do recognize that such change would cause some confusions and will make a note in future publications. As the reviewer noted, many of our models have been involved in several completed projects and publications. The revised figure legend is shown below:
Figure 3. Representative images of IHC staining of PDOX tumors with human specific antibodies against mitochondria (A) and vimentin (B). Tumor (Tum), positive stained tumor cells (arrow) and blood vessels (Bv) are indicated. Note: pediatric GBM is replace by high-grade glioma (of different subtypes) in WHO CNS Tumor Classification (2021). Bar=100 µM
In Figure 3, Ki-67 panel is missing.
Response: It was a copy and paste error. We have removed the Ki-67.
Missing scale bar in Figure 3.
Response: Yes, added.
In figure 4 C, Viable cell (PI-), second panel annotation is confusing (Viable cell PI-)\
Response: We added the mouse-specific antibodies to the figure legend as below to help clarify the assay:
- C. Flow cytometric analysis of human and mouse cells using human specific antibodies against HLA-ABC and an antibody cocktail (CD19, CD133, CD40 and CD140) specific to mouse brain cells.
Line 647 (Heading)- Typo (dRug)
Response: Corrected.
Line 648-651- If possible, authors can simplify this statement.
Response: Yes, the statement is simplified as below:
- Due to the invasive growth of PDOX tumor in mouse brain, mouse cells are frequently present in the dissociated xenograft tumor cells. Therefore, determination of the cell origin (mouse or human) should be examined periodically either with RT-PCR using human-specific and mouse specific-genes or FCM or both.
Author’s self-citation looks self-aggrandizement. I may suggest them to look for similar articles from other research groups as well to be propose more generalized protocol for the benefit of readers.
Response: Our purpose of self-citation on this protocol paper is to show the readers that we have tested and used our protocols in multiple tumor types. We thought this is important to convince the readers that this is a well-developed protocol. Listing our publications is a humble presentation to the readers of our work, although we have to admit that we are slightly proud of what we have achieved with our model development protocol.
Round 2
Reviewer 2 Report (New Reviewer)
Comments and Suggestions for Authors
The authors have revised the work, included their comments on the concerns and changed figures accordingly in present article. I have no additional suggestions.
Author Response
Thank you very much for your review.
This manuscript is a resubmission of an earlier submission. The following is a list of the peer review reports and author responses from that submission.
Round 1
Reviewer 1 Report
Comments and Suggestions for Authors Thank you for the opportunity to read this work. My main problem with this work is that it does not really benefit the existing literature. The technique of free hand injection of tumor cells has been described, including by the authors of this paper. One benefit of this work would be to find out the number of incorrect injections. This would allow a comparison with stereotactic injections. However, the authors only mention the number of cases in which tumor growth occurred. At the same time, there are some problems with the content of the manuscript, some of which I list in the appendix. To summarize, I do not think that this paper should be published in its current state, as it does not bring up any relevant new aspects. Methods: In our laboratory it is common practice to use trypsin to prepare a single cell suspension. However, this is apparently not necessary in your case. Perhaps this should also be mentioned in the methods section. Methods. "Make a small (1 mm) skin incision to the right of the sagital sinus and 2 mm anterior 240 (for intra-cerebral injections) (Fig. 2e) or 1 mm posterior (for intra-cerebellar injections) (Fig. 2i) to the bregma occipital line or at the upper right corner between the sagital sinus and the bregma occipital line (Fig. 1m)" > Please give a number in mm for how far to the right parasagittal you have to go. > Please indicate how you define / touch the bregma occipital line, since all procedures base on that anatomical structure. > Please give a short statement why you use the right and not the left side. > Where do you make the burr hole for intra-brainstem injections? Methods. Blood vessel on the cranial surface should be avoided, particularly for intra-cerebellar injections. > How can you avoid this looking through a 0.7mm burr hole? Or do you mean extra-cranial blood vessels, then this remark should be noted in the section dealing with skin incision. Bleeding was not commonly observed with intra-cerebral injection; less than 50 µL of blood loss can be seen with intra-cerebellar injection. > Please explain this phenomenon. Methods. How many seconds do you take for the injection? Methods. "After surgery, all the animals will be kept warm, clean and dry throughout the immediate post-operative period" > Why is postoperative pain medication not necessary here? Figure 1. There is a discrepancy between the illustration and the text. In the illustration, 1ul volumes are selected for brainstem electrical activity, in the text 2ul volumes are mentioned. This is a problem. Figure 1. You have C but there is no figure legend for that part of the figure. Unfortunately, there are more and more errors in the manuscript, which show that not much effort was put into the writing. Figure 1. Please explain the histology / tumor entity of the cell lines you used. Results. "Following this procedure, we have successfully developed 185 xenograft models from 407 pediatric CNS surgical samples, with a success rate of 72.3% in GBM, 64.2% in medulloblastoma, 50% in ATRT, 33.8% in ependymoma and 11.6% in low grade gliomas" > Cell lines a/o patient derived tumor cells? > Why do you cite your own reference (39) although this study does not proof your assumption here? Results. I think that the whole process after brain removal is superfluous. This is solely about the topic of free hand implantation, all other process steps seem somewhat out of place here.Author Response
Please see the attachment.

Reviewer 2 Report
Comments and Suggestions for Authors
The manuscript describes a protocol for free-hand transplantation of brain tumor surgical specimens into specific locations in the brains of SCID mice. The procedures include preparation of fresh patient tumor tissue, the actual transplantation into mice brains, follow-up monitoring, troubleshooting, and long-term cryopreservation of xenograft cells. Additionally, it discusses the initiation of cell cultures and statistical considerations for pre-clinical drug testing. Overall, it presents a detailed and efficient protocol for free-hand transplantation of brain tumor surgical specimens into mouse brains as a labor-efficient and reproducible alternative to traditional stereotaxic frame methods. However, there are some questions still needed to be answered:
1) In addition to transplanting classic monolayer cells, have the authors investigated the transplantation of neurospheres into the mouse brain? It would be interesting to check any potential differences in engraftment, growth characteristics, and tumor development between classic monolayer cells and neurospheres.
2) Have the authors thought to extend their experimentation to include transplantation studies in pups or newborn mice?
3) While the use of a stereotaxic frame ensures precision and consistency in each injection, the manuscript should address the potential impact of free-hand transplantation on inducing variation.
Minors:
1. Some extra space among texts.
2. Line 196, should be 100 um and 40 um strainers.
Reviewer 3 Report
Comments and Suggestions for Authors
-
Dear Authors,
-
I wanted to express my sincere appreciation for your outstanding work in the article titled "Direct Implantation of Patient Brain Tumor Cells into Matching Locations in Mouse Brains for Patient-Derived Orthotopic Xenograft Model Development." Your research is of paramount importance in the field of neurooncology, addressing a critical clinical issue.
Here are a few words of commendation:
-
Innovative Technique: Your development of a surgical technique for the rapid and precise implantation of patient brain tumor samples into mouse brains is truly remarkable. Its simplicity and safety, along with the impressive success rate, make it a valuable tool for advancing research in this area.
-
Impactful Results: The creation of 185 PDOX models from a substantial number of patient samples demonstrates the practicality and efficacy of your method. Your ability to faithfully replicate the histopathological and genetic characteristics of the original tumors is commendable.
-
Potential for Advancement: Your work not only simplifies laboratory research but also holds great promise for accelerating the development of effective therapies for brain tumor patients. It has the potential to make a significant impact on clinical practice.
-
Contribution to Science: Your article provides a valuable contribution to the scientific community by improving our understanding of brain tumors and offering a practical approach to studying them in a preclinical setting.
I want to extend my heartfelt congratulations to you on this achievement. Your dedication and innovation are vital in advancing medical research and improving the lives of patients. I encourage you to continue your valuable work in this field, as it has the potential to bring about transformative changes in the treatment of brain tumors.
Once again, congratulations on your exceptional research, and I look forward to seeing your work published and further recognized in the scientific community. In summary, the article is highly valuable in the field of neurooncology research and may contribute to finding new therapies for brain tumor patients. The authors deserve congratulations for developing and refining this technique and demonstrating its effectiveness in creating PDOX models. This is an important contribution to the field of medicine that deserves publication.
